# Mechanistic insights into chromatin targeting by leukemic NUP98-PHF23 fusion

Yi Zhang [1], Yiran Guo [2], Sheryl M. Gough[3], Jinyong Zhang[1], Kendra R. Vann [1], Kuai Li[4], Ling Cai[2], Xiaobing Shi [4], Peter D. Aplan[3], Gang Greg Wang [2] & Tatiana G. Kutateladze [1]✉

Chromosomal NUP98-PHF23 translocation is associated with an aggressive form of acute myeloid leukemia (AML) and poor survival rate. Here, we report the molecular mechanisms by which NUP98-PHF23 recognizes the histone mark H3K4me3 and is inhibited by small molecule compounds, including disulfiram that directly targets the PHD finger of PHF23 (PHF23PHD). Our data support a critical role for the PHD fingers of NUP98-PHF23, and related NUP98-KDM5A and NUP98-BPTF fusions in driving leukemogenesis, and demonstrate that blocking this interaction in NUP98-PHF23 expressing AML cells leads to cell death through necrotic and late apoptosis pathways. An overlap of NUP98-KDM5A oncoprotein binding sites and H3K4me3-positive loci at the Hoxa/b gene clusters and Meis1 in ChIP-seq, together with NMR analysis of the H3K4me3-binding sites of the PHD fingers from PHF23, KDM5A and BPTF, suggests a common PHD finger-dependent mechanism that promotes leukemogenesis by this type of NUP98 fusions. Our findings highlight the direct correlation between the abilities of NUP98-PHD finger fusion chimeras to associate with H3K4me3-enriched chromatin and leukemic transformation.

[1] Department of Pharmacology, University of Colorado School of Medicine, Aurora, CO 80045, USA. [2] Department of Biochemistry and Biophysics, Curriculum in Genetics and Molecular Biology, Lineberger Comprehensive Cancer Center, The University of North Carolina School of Medicine, Chapel Hill, NC 27599, USA. [3] Genetics Branch, Center for Cancer Research, National Cancer Institute, NIH, Bethesda, MD 20892, USA. [4] Center for Epigenetics, Van Andel Research Institute, Grand Rapids, MI 49503, USA. ✉email: tatiana.kutateladze@cuanschutz.edu

Posttranslational modifications (PTMs) of histones provide fundamental mechanisms for modulating gene expression and determining cellular identities during organismal development and cell lineage specification. Misregulation of these mechanisms is associated with a number of human diseases, including acute myeloid leukemia (AML)[1]. AML is a malignancy of myeloid precursor cells, which is characterized by uncontrolled cell proliferation and arrested terminal differentiation. Clinical studies have identified a direct link between the development of AML and aberrant activities of proteins that play pivotal roles in epigenetic processes through installing, removing, or binding histone PTMs. For instance, chromosomal rearrangements that produce fusion genes involving the histone H3 lysine 4 (H3K4)-specific methyltransferase MLL1 account for ~70% of infant leukemias and ~5–10% of childhood and adult AML cases[2]. A subset of AML cases shows abnormal chromosomal translocations that involve nucleoporin 98 (NUP98) and its fusion partners and components of the epigenetic machinery, such as KDM5A/JARID1A, NSD1, NSD3, LEDGF, BPTF, and PHF23[3–9]. The fused chimeras activate *HOX* gene-dependent oncogenic programs and induce leukemic transformation in mice and in human hematopoietic stem cells[4,8–10].

NUP98, a component of the nuclear pore complex, was found on nucleoplasmic and cytoplasmic domains of the complex and in the nuclear interior. The nuclear pool of NUP98 has been shown to associate with spherical speckles—the GLFG (glycine, leucine, phenylalanine, and glycine) bodies—and FRAP (fluorescence recovery after photobleaching) experiments reveal that NUP98 is highly mobile within the nucleus of living cells[11]. NUP98 contains multiple GLFG repeats in its N-terminus that are required for the association of NUP98 with the GLFG bodies (hence the name of the bodies)[11], and the mobility of NUP98 is strongly coupled to ongoing transcription[11,12]. Recent structural analysis of the NUP98 GLFG repeat shows that the repeat forms kinked β sheets with the aromatic side chain of phenylalanine being involved in intra- and inter-sheet stabilization, whereby providing a mechanism for a low complexity aromatic-rich kinked segments (LARKS)-dependent oligomerization[13].

The NUP98–PHF23 fusion was originally identified as a cryptic translocation in AML in 2007[14], and like other NUP98 fusion chimeras, NUP98–PHF23 translocation is associated with an aggressive disease course and poor survival[15]. We have previously shown that the expression of NUP98–PHF23 fusion leads to myeloid, erythroid, T-cell, and B-cell leukemia in mice, and that the treatment of NUP98–PHF23 cells with a small molecule compound disulfiram (DS) results in death of cells expressing NUP98–PHF23, likely due to disruption of the histone-binding activity of the PHF23 plant homeodomain (PHD) finger[9]. Despite the critical role of NUP98–PHF23 in leukemogenesis, which emphasizes the urgent need to develop chemotypes targeting this chimera, the structure–function relationship of PHF23 remains poorly characterized.

In this study, we report molecular mechanisms for the recognition of methylated histone H3K4 by the PHD finger of PHF23 and inhibition of this interaction by small molecule compounds DS and amiodarone (AD). Our data corroborate the critical role of the PHD finger of NUP98–PHF23 in driving leukemogenesis and demonstrate that blocking this interaction leads to cell death through necrotic and late apoptosis pathways. The NMR-derived mapping of binding interfaces suggests the mechanism by which DS acts on NUP98–PHF23 and on related NUP98–KDM5A and NUP98–BPTF fusions. Our results shed further light on the direct correlation between the capabilities of NUP98 fusion chimeras to associate with H3K4me3-enriched chromatin and leukemogenesis.

## Results and discussion

**The PHD finger of PHF23 selects for H3K4me3.** In the NUP98–PHF23 chimera, the amino-terminal portion of NUP98 with multiple GLFG repeats is fused to the carboxy-terminal portion of PHF23 that contains a single PHD finger (PHF23$_{PHD}$) (Fig. 1a, b). Although PHF23$_{PHD}$ was shown to associate with biotinylated H3K4me3 peptide in a pulldown experiment[8], neither selectivity of this domain for histone H3 nor the binding mechanism is known. To determine preference of PHF23$_{PHD}$ for the H3K4 methylation state, we expressed and purified $^{15}$N-labeled PHF23$_{PHD}$ and examined its interactions with H3K4me3, H3K4me2, H3K4me1, and H3K4me0 peptides by $^{1}$H,$^{15}$N heteronuclear single quantum coherence (HSQC) experiments (Fig. 1c). Titration of the H3K4me3 peptide (aa 1–12 of H3K4me3) into the NMR sample caused substantial chemical shift perturbations (CSP) in PHF23$_{PHD}$. In addition to CSPs, a number of cross-peaks of the protein broadened and disappeared, indicating tight binding in intermediate exchange regime on the NMR time scale. Titration of H3K4me2 peptide led to an almost identical pattern of CSP, and the intermediate exchange regime again pointed to a strong interaction and binding affinity in a low micromolar range. In agreement, dissociation constants ($K_d$s) for the complexes of PHF23$_{PHD}$ with the H3K4me3 and H3K4me2 peptides were found to be 2 and 8 μM, respectively, as measured by intrinsic tryptophan fluorescence (Fig. 1d, e). In contrast, H3K4me1 and H3K4me0 peptides induced CSP in PHF23$_{PHD}$ in fast exchange regime, and plotting these resonance changes against peptide concentration yielded $K_d$s of 580 μM and 1.6 mM, respectively. These results suggest that PHF23$_{PHD}$ selects for the higher methylation state of H3K4 (tri- and dimethylated) and discriminates against monomethylated (H3K4me1) or unmodified H3. We note that the binding affinity of PHF23$_{PHD}$ toward H3K4me3/2 is in the range of binding affinities exhibited by the majority of epigenetic readers that bind to histone tails (1–50 μM)[16–18]. High selectivity of PHF23$_{PHD}$ toward H3K4me3 was corroborated by peptide pulldown assays (Fig. 1f). GST-tagged PHF23$_{PHD}$ was incubated with biotinylated histone H3 and H4 peptides either unmodified or containing mono, di- and tri-methylation marks commonly found in chromatin. We found that PHF23$_{PHD}$ recognizes methylated Lys4 of H3 but does not associate with any other methylation marks of H3 or H4 tested and that symmetric or asymmetric dimethylation of Arg2 does not affect binding of PHF23$_{PHD}$ to H3K4me3.

**The aromatic cage of PHF23$_{PHD}$ can accommodate lysine mimetics.** What is the mechanism underlying leukemic activity of NUP98–PHF23? A fast (~20 s) recovery after photobleaching of the NUP98 GLFG bodies[11] suggests that the GLFG bodies have a liquid/gel like property. Moreover, NUP98–PHF23 fusion is also capable of associating with nuclear speckles though smaller in size[19], and LARKS-dependent oligomerization is known to yield a liquid phase[13,20,21]. In the case of NUP98–PHF23, formation of the phase separated nuclear bodies/condensates through GLFG repeats oligomerization would result in high (~100-fold higher)[13,20,21] concentration of the protein at certain genomic sites, furthering binding of PHF23$_{PHD}$ to H3K4me3 and causing aberrant transcriptional activation (Fig. 2a). To better understand the mechanism by which PHF23$_{PHD}$ recognizes the histone tail, we determined the protein structure. Although PHF23$_{PHD}$ was crystallized in the presence of H3K4me3 peptide, unexpectedly we obtained the structure in which the N-terminal sequence of one PHF23$_{PHD}$ molecule was bound to the second PHF23$_{PHD}$ molecule (Fig. 2b, Supplementary Fig. 1, and Supplementary Table 1). Alignment of this structure with the structures of other PHD fingers, such as of

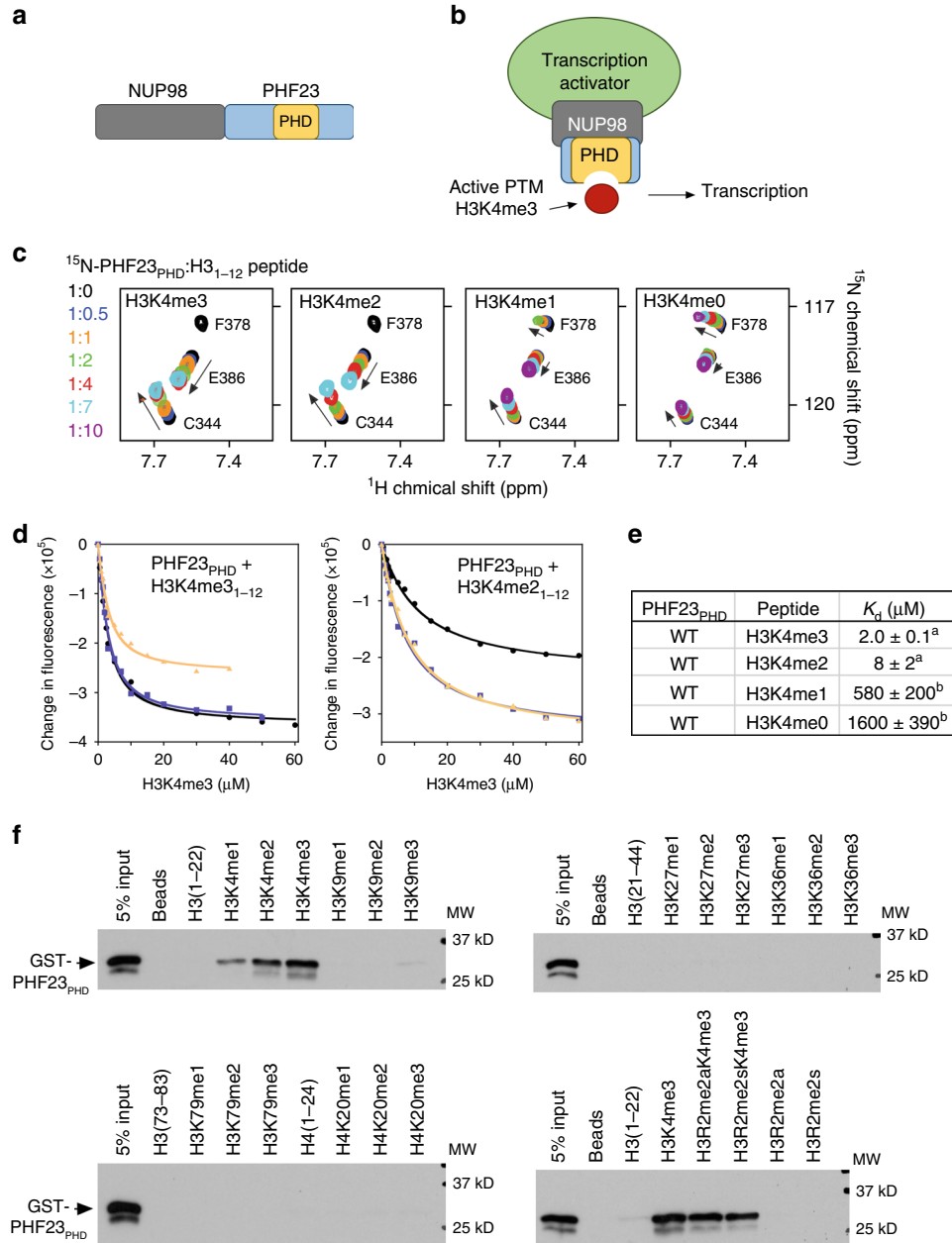

**Fig. 1 PHF23_PHD binds H3K4me3. a** Schematic of the NUP98–PHF23 fusion protein. **b** A model for NUP98–PHF23 dependent leukemic transformation. Binding of the PHD finger of PHF23 to H3K4me3, a mark associated with active gene transcription, bridges transcription activators and/or prevents binding of transcription silencing components. **c** Superimposed $^1$H,$^{15}$N HSQC spectra of PHF23_PHD collected upon titration with indicated H3 peptides. Spectra are color coded according to the protein:peptide molar ratio. **d** Binding curves used to determine the $K_d$ values by fluorescence spectroscopy. Data points and fitted curves for three independent experiments are shown and colored black, blue, and wheat. Source data are provided in the Source Data file. **e** Binding affinities of WT PHF23_PHD for the indicated histone peptides measured by tryptophan fluorescence ([a]) or by NMR titration experiments ([b]). Error represents SD in triplicate measurements. **f** Western blot analysis of peptide pulldowns of GST-PHF23_PHD with the indicated histone H3 peptides from single experiment. Source data are provided in the Source Data file.

KDM5A and BPTF in complex with H3K4me3 peptides (see discussion below), revealed that the N-terminal sequence DLIT (residues 338–341 of PHF23_PHD) and a preceding serine derived from the vector superimpose very well with H3K4me3 in the KDM5A and BPTF complexes. Much like H3K4me3, the SDLIT sequence is bound in an extended conformation and pairs with the β1 strand of PHF23_PHD, which results in the formation of the three-stranded antiparallel β-sheet. The I340 side chain occupies the aromatic/hydrophobic cage of PHF23_PHD consisting of F348, M353, and W362 with another hydrophobic residue, L339 being positioned to make an additional wall of the cage.

To gain insight into the interaction with H3K4me3 peptide, we assigned resonances of the protein backbone amides through collecting and analyzing three-dimensional triple resonance NMR spectra of uniformly $^{13}$C,$^{15}$N-labeled PHF23_PHD (Supplementary Fig. 2). Mapping the most perturbed residues of the protein (due to binding of H3K4me3) on the surface of PHF23_PHD confirmed that H3K4me3 occupies the same binding pocket of PHF23_PHD as the SDLIT sequence (Fig. 2c, d). The importance of the aromatic cage residues was confirmed by mutagenesis. Substitution of either M353 with valine or W362 with alanine abolished the interaction with H3K4me3, and replacement of F348 with

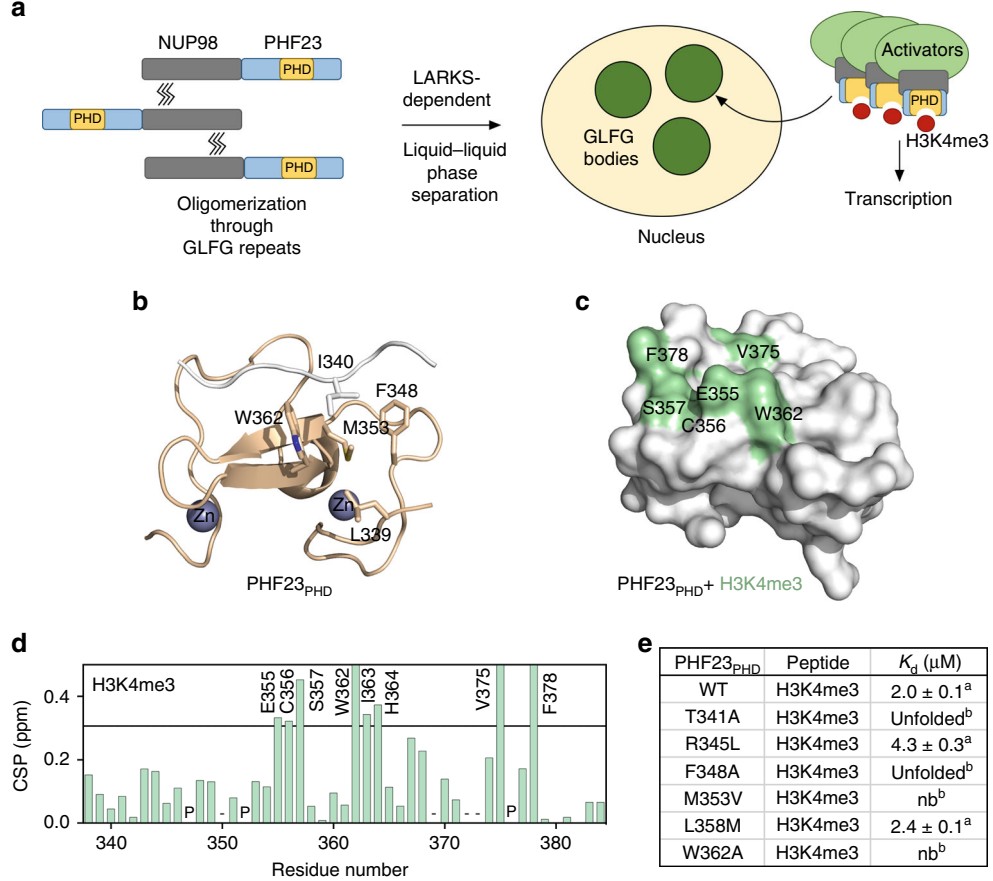

**Fig. 2 Molecular basis for the recognition of H3K4me3 by PHF23$_{PHD}$. a** A model for the leukemic activity of the NUP98–PHF23 fusion. **b** Structure of PHF23$_{PHD}$. The N-terminus of another PHF23$_{PHD}$ molecule (white) binds in the histone-binding site of the domain. The side chain of I340 occupies the aromatic cage. Source data are provided in the Source Data file. **c, d** Identification of the H3K4me3-binding site of PHF23$_{PHD}$. Residues that exhibit H3K4me3-induced resonance changes in (**d**) are mapped onto the structure of PHF23$_{PHD}$ in (**c**). Histogram shows NMR chemical shift perturbations in PHF23$_{PHD}$ upon binding of H3K4me3 peptide at a 1:4 ratio. 'P' indicates a proline residue. '−' indicates an unassigned residue. Bars reaching the maximum of *y*-axis indicate disappeared cross-peaks. Source data are provided in the Source Data file. **e** Binding affinities of the PHF23$_{PHD}$ mutants to the H3K4me3 peptide as measured by tryptophan fluorescence ([a]) or by NMR titration experiments ([b]).

alanine led to protein unfolding, suggesting a role of this residue in structural stability of PHF23$_{PHD}$ (Fig. 2e). Although PHF23$_{PHD}$ recognizes higher methylation states of H3K4 and interacts substantially weaker with H3K4me1 or unmodified H3K4, similar directional patterns of CSPs observed upon binding of each peptide imply that the aromatic cage can accommodate either methylated or unmodified lysine (Supplementary Fig. 3). Of note, I340 in the SDLIT sequence superimposes well with the hydrophobic part of the K4me3 side chain, further suggesting that lysine mimetics could also occupy the aromatic cage (Fig. 2b).

**Amiodarone (AD) and disulfiram (DS) directly bind to PHF23$_{PHD}$.** Several small molecule compounds that target the PHD finger in NUP98–PHF23 and show promising results in inhibiting leukemogenesis have been identified, including the FDA approved antiarrhythmic drug AD and DS used for the treatment of chronic alcoholism, however, their mechanisms of action remain unclear[9,22] (Fig. 3a). Furthermore, fluorescence assays reveal that the cancer-relevant mutations R345L and L358M (Cosmic) (both R345 and L358 are located far from the H3K4me3-binding site) do not disrupt binding of PHF23$_{PHD}$ to H3K4me3, reinforcing the idea that inhibition of the H3K4me3-PHF23$_{PHD}$ interaction might be beneficial in destroying cancer cells (Fig. 2d). To better understand the mechanism by which AD and DS act on NUP98–PHF23, we assayed these compounds by NMR titration experiments. Gradual

addition of AD or DS to PHF23$_{PHD}$ induced small but significant CSPs in the protein, indicative of direct interactions (Fig. 3b, c). These results were in agreement with the reports demonstrating that AD and its analogues display inhibitory activity toward H3K4me3-binding domains, whereas some PHD fingers are sensitive to DS[22]. Treating a NUP98–PHF23 expressing cell line (748T) vs. control cell line (7298/2) with up to 2.2 μM AD for 96 h showed modest inhibition of 748T cell survival, however, treatment of a myeloid NUP98–PHF23 expressing (961C) cell line vs. control (189E6 and 32D) for 48 h with 2 μM DS showed robust inhibition (Fig. 3d–f), corroborating our previous findings[9]. Together, NMR and cell data suggest that while both compounds directly target PHF23$_{PHD}$, DS elicits a much stronger effect than AD in NUP98–PHF23 expressing leukemia cells.

**AD and DS mechanisms of action differ.** To elucidate the molecular mechanisms underlying inhibitory effects of AD and DS, we plotted CSPs observed in $^{1}$H,$^{15}$N HSQC spectra of PHF23$_{PHD}$ upon titration of the compounds (Fig. 3b, c) onto the protein surface and in parallel examined the reaction mixtures by electrophoresis. Close evaluation of the NMR titration experiments revealed that the most affected residues of PHF23$_{PHD}$ were located in and around its H3K4me3-binding pocket, implying that both compounds associate with PHF23$_{PHD}$ near the binding site, at least at the protein:compound ratio of 1:20 for AD and 1:2 for DS

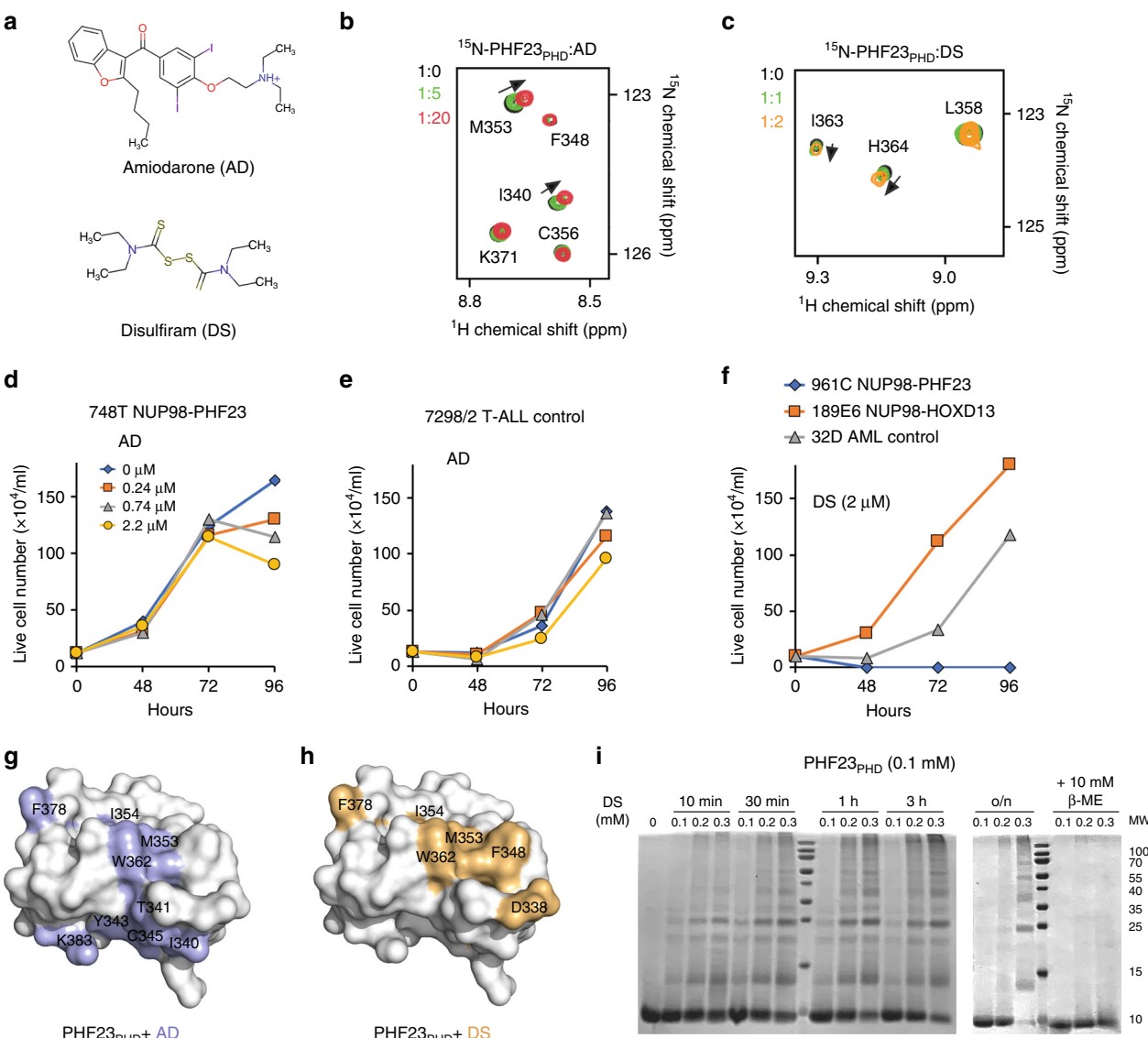

**Fig. 3 AD and DS bind to PHF23_PHD and inhibit the survival of NUP98–PHF23 expressing AML cells. a** Chemical structures of amiodarone and disulfiram. **b**, **c** Superimposed $^1H,^{15}N$ HSQC spectra of PHF23_PHD collected upon titration with AD or DS. Protein concentration was kept at 100 μM. Concentrations of AD were 0, 0.5, and 2 mM **b**. Concentrations of DS were 0, 0.1, and 0.2 mM **c**. Spectra are color coded according to the protein: compound molar ratio. **d**, **e** AD treatment affected the survival of NUP98–PHF23 (748T) and control (7298/2) cell lines. For additional trials see ref. [9]. Source data are provided in the Source Data file. **f** Concentration dependent effect of DS on the survival of NUP98–PHF23 expressing 961C, NUP98–HOXD13 expressing 189E6, and 32D AML cell lines. For additional trials see ref. [9]. Source data are provided in the Source Data file. Identification of the AD-binding **g** and DS-binding **h** sites of PHF23_PHD. Residues that exhibit significant ligand-induced resonance perturbations in **b**, **c** are mapped onto the structure of PHF23_PHD. See also Supplementary Fig. 4. **i** Samples containing 0.1 mM PHF23_PHD were incubated with indicated amounts of DS for 10 min, 30 min, 1 h, 3 h, and overnight (o/n). All samples were flash-frozen and resolved by SDS-PAGE under nonreducing and reducing (10 mM mercaptoethanol (β-ME)) conditions. Experiment was repeated independently two times with similar results.

(Fig. 3g, h and Supplementary Fig. 4). Addition of excess DS (ratio of 1:3) or AD (ratio of 1:50) led to disappearance of amide resonances of PHF23_PHD, indicating the formation of high molecular weight species that exceed detection limit of NMR. Subsequent analysis of the reaction mixtures by SDS-PAGE gel electrophoresis revealed distinct modes of action for these compounds. We observed a single band corresponding to the size of a monomeric protein in the PHF23_PHD:AD sample resolved on a nonreducing SDS-PAGE gel (Supplementary Fig. 5). We concluded that at high concentrations, AD stimulates non-covalent oligomerization of PHF23_PHD. In contrast, in the presence of DS, nonreducing SDS-PAGE gels showed a ladder of protein oligomers, clearly observed after 30 min incubation of PHF23_PHD with DS at a protein:

compound ratio of 1:3 (Fig. 3i). Markedly, the PHF23_PHD ladder bands disappeared upon addition of the reducing agent, β-mercaptoethanol (β-ME). These results suggest that at high concentration and over time DS reacts with zinc chelating Cys residues of PHF23_PHD generating covalently linked intermolecular oligomers (Fig. 4a). Although we cannot exclude the formation of the intramolecular intermediates, in either case the DS activity would result in eventual and permanent disruption of the PHF23_PHD fold.

## DS promotes NUP98–PHF23 fusion cell death through necrosis and late apoptosis.

We previously used chromatin immunoprecipitation-sequencing (ChIP-seq) to demonstrate that a NUP98–PHF23 fusion protein colocalizes with H3K4me3 at a

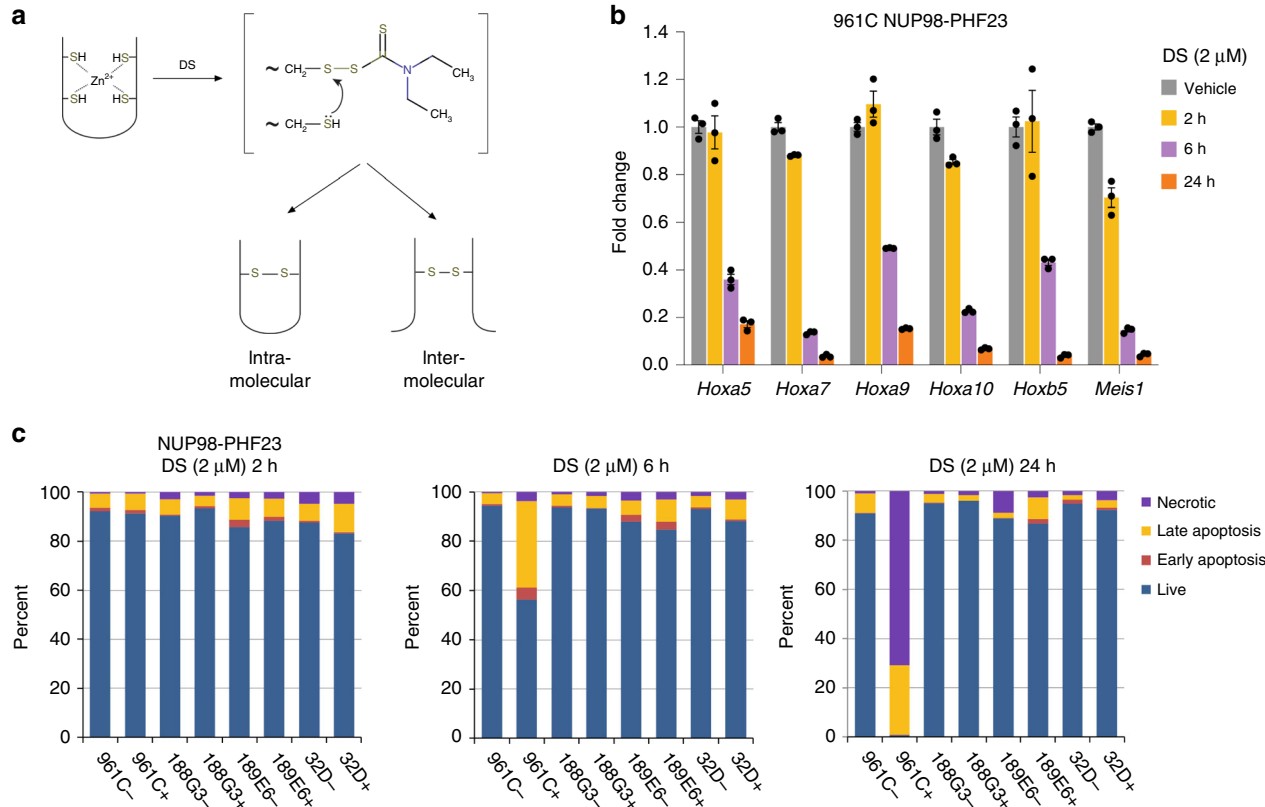

**Fig. 4 DS induces cell death in NUP98–PHF23 expressing cells. a** The mechanism of DS-based inhibition. **b** DS treatment leads to reduced target gene expression. qRT-PCR quantification of previously identified[9] target genes normalized to 18S RNA as internal control confirms downregulation of Hoxa5, Hoxa7, Hoxa9, Hoxa10, Hoxb5, and Meis1 in the presence of DS. Error bars represent SEM ($n = 3$ each sample). Source data are provided in the Source Data file. **c** Proportions of necrotic, late apoptosis, early apoptosis, and live cells in NUP98–PHF23 expressing 961C, NUP98–HOXD13 expressing 188G3 and 189E6, and control (32D) AML cell lines treated without (−) or with (+) DS. $n = 1$, source data are provided in the Source Data file.

limited number of target sites, including Hoxa7, Hoxa9, Meis1, and Hoxb5; genes that are known to be important for hemato-poietic stem and precursor cell function[9]. DS treatment of 961C cells that express a NUP98–PHF23 fusion showed rapid and profound downregulation of target gene (Hoxa5, Hoxa7, Hoxa9, Hoxa10, Hoxb5, and Meis1) mRNA expression within 6 h of treatment with 2 μM DS (Fig. 4b). Downregulation of these genes was accompanied by rapid death (>95% at 24 h through necrotic and late apoptosis pathways) of the NUP98–PHF23 expressing cells, but not of three control cell lines, including 188G3, 189E6, and 32D (Fig. 4c), underscoring again the selective targeting of the PHD finger in the NUP98–PHF23 fusion by DS.

**Functional PHD finger is critical in leukemic transformation.** Much like NUP98–PHF23, fusions of NUP98 with BPTF and KDM5A contain a single PHD finger in the C-terminal part of the chimeras (Fig. 5a). As shown in Fig. 5b, all three fusions efficiently induced immortalization and transformation of murine hemato-poietic stem/progenitor cells (HSPCs). To assess the mechanistic role of the PHD fingers in these processes, we compared the structure of PHF23$_{PHD}$ with that of KDM5A$_{PHD}$ (the third PHD finger of KDM5A)[8] and BPTF$_{PHD}$[23]. The aromatic cage of BPTF$_{PHD}$ appears to be the most complete, consisting of four aromatic residues that fully enclose the trimethylammonium group of lysine 4, whereas the aromatic cages of PHF23$_{PHD}$ and KDM5A$_{PHD}$, although having different topologies, both contain two aromatic residues (Fig. 5c, d). Despite the differences in aromatic cage architecture, the three domains exhibit similar binding affinities to H3K4me3 ($K_{d}s = 3$, 2, and 0.9 μM for BPTF[23], PHF23, and KDM5A[8], respectively). To

determine whether the PHD–H3K4me3 interaction per se is responsible for the leukemic transformations (Fig. 5b), we generated NUP98–BPTF chimeras harboring Y10A or W23A mutations that have been shown to abrogate BPTF$_{PHD}$ interaction with H3K4me3[23]. In contrast to wild-type NUP98–BPTF fusion, the mutant NUP98–BPTF Y10A or NUP98–BPTF W23A fusions failed to induce the oncogenic transformation of hematopoietic stem/progenitors (Fig. 5e, f). These data together with previous findings that mutations in the aromatic cages of NUP98–KDM5A and NUP98–PHF23 abrogate their transforming ability[8], point to the universal requirement of a functional PHD finger for leuke-mogenesis in this type of oncogenic chimeras.

**NUP98–KDM5A is enriched at H3K4me3 regions and is sensitive to DS.** We next carried out ChIP-seq experiments to map the genomic occupancy of NUP98–KDM5A as well as the histone methylation marks H3K4me3 and H3K27me3 in the AML cells (Fig. 6a, b and Supplementary Fig. 6a). We found that there is a strong overlap of NUP98–KDM5A oncoprotein binding sites and H3K4me3-positive loci (Supplementary Fig. 6b), as exemplified by the Hoxa/b gene clusters and Meis1 (Fig. 6a). However, H3K27me3, a gene repressive histone mark, was found to be absent from these AML-related genes (Fig. 6b), a pattern that we also observed in AML cells carrying NUP98–PHF23[9]. Collectively, our data indicate an overall similar mode of action for these NUP98 fusions during AML development that depends on binding of their PHD fingers to H3K4me3, a mark enriched at promoters of actively transcribed genes. These results also suggest that the PHD finger-targeting chemotypes could inhibit the

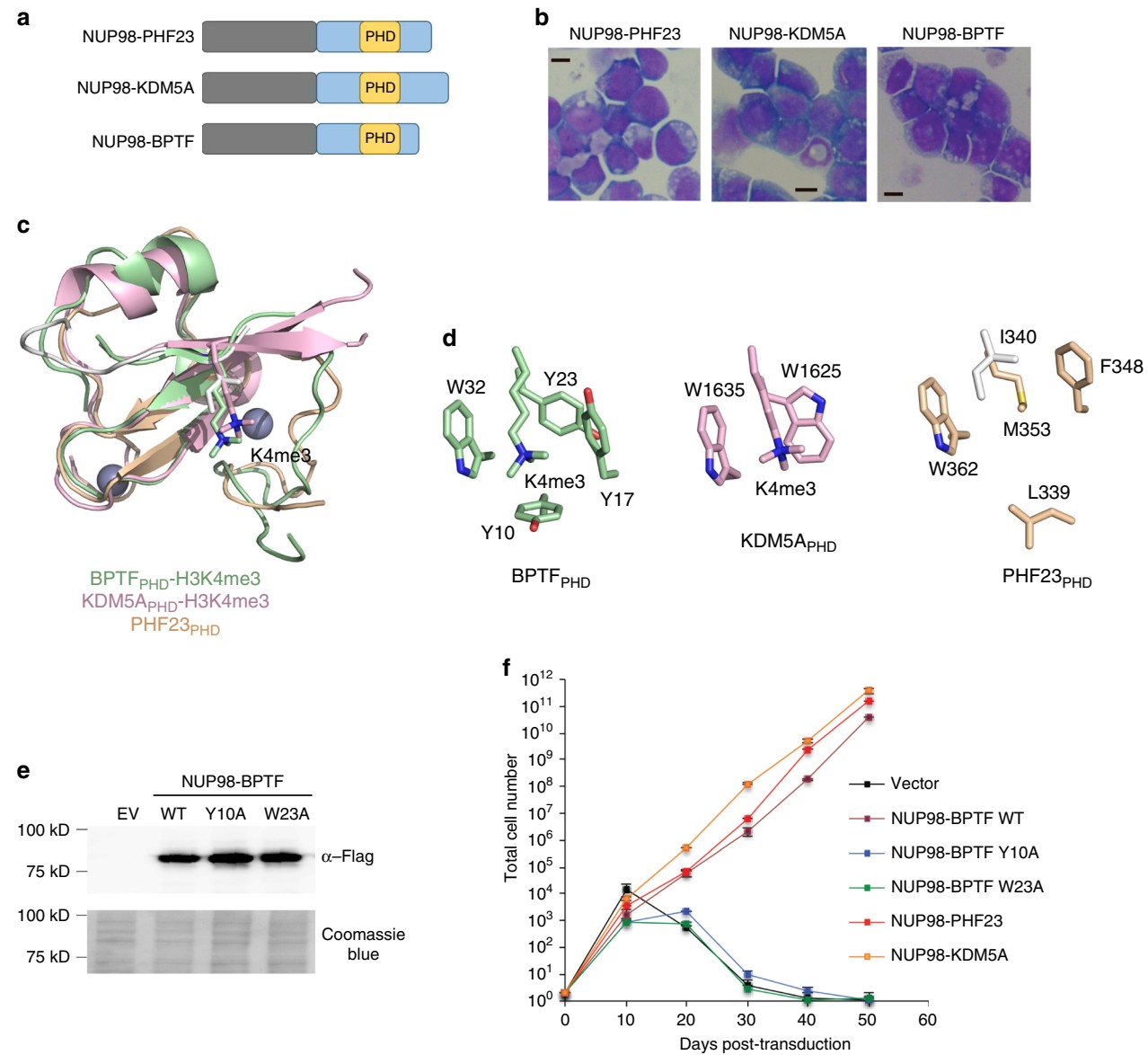

**Fig. 5 PHD finger-dependent leukemic transformation. a** Schematic representation of NUP98 fusion proteins identified in AML cases. **b** Wright–Giemsa staining of murine hematopoietic progenitor cells transformed by the indicated NUP98 fusion proteins. Scale bars: 5 µM. Images were from single experiment. **c** Superimposed structures of the BPTF$_{PHD}$ (ref.[23]), KDM5A$_{PHD}$ (ref.[8]), and PHF23$_{PHD}$ in complex with indicated ligands. PDB IDs: 2F6J, 3GL6, and 6WXK. The structure of PHF23$_{PHD}$ superimposes with the structures of KDM5A$_{PHD}$ and BPTF$_{PHD}$ with a RMSD of 0.68 and 1.32 Å, respectively. **d** A zoom-in view of the aromatic cages of BPTF$_{PHD}$, KDM5A$_{PHD}$, and PHF23$_{PHD}$. **e** FLAG immunoblots from single experiment showing protein levels of the indicated FLAG-tagged NUP98 fusion post-transduction into hematopoietic progenitor cells. **f** Proliferation of the indicated NUP98 fusion post-transduction into hematopoietic progenitor cells. Error represents SD of triplicate measurements, $n = 3$ biologically independent experiments. Source data are provided in the Source Data file.

NUP98 fusions in unison. Indeed, the treatment of cells expressing NUP98–KDM5A with DS showed DS concentration dependent cell death, in which 1 µM DS treatment led to <~1% live cells after 24 h (Fig. 6c).

**Conservation of the PHD finger-dependent inhibition**. To gain mechanistic insights into PHD finger-dependent inhibition, we investigated the effect of DS and AD on KDM5A$_{PHD}$ and BPTF$_{PHD}$ by NMR and electrophoresis (Supplementary Figs. 7 and 8). As in the case of PHF23$_{PHD}$, significant CSPs were observed in HSQC spectra of $^{15}$N-labeled KDM5A$_{PHD}$ and BPTF$_{PHD}$ upon addition of DS, indicating binding of the compound (Fig. 6d, e). In contrast to the uniform disappearance of NMR signals observed for PHF23$_{PHD}$ at a protein:compound ratio of 1:3, resonances of KDM5A$_{PHD}$ and

BPTF$_{PHD}$ were still visible at this ratio and, additionally, a set of new cross-peaks emerged in the middle of the spectrum between 7.5 and 8.5 ppm in the $^1$H dimension, likely representing resonances of unfolded protein conformations. In agreement, SDS-PAGE gel assays showed formation of disulfide-bonded oligomers in BPTF$_{PHD}$ and KDM5A$_{PHD}$, implying that DS interacts with these PHD fingers through a conserved mechanism via generating cysteine-dependent disulfide species (Supplementary Fig. 8). We note that in addition to zinc-coordinating Cys residues, KDM5A$_{PHD}$ contains an extra free cysteine, which likely contributes to the formation of primarily a dimer. Nevertheless, our data suggest that DS is capable of binding to PHD fingers and reacting with their either free or zinc-coordinating Cys residues, disrupting the structures and therefore histone-binding activities of all three NUP98 fusions.

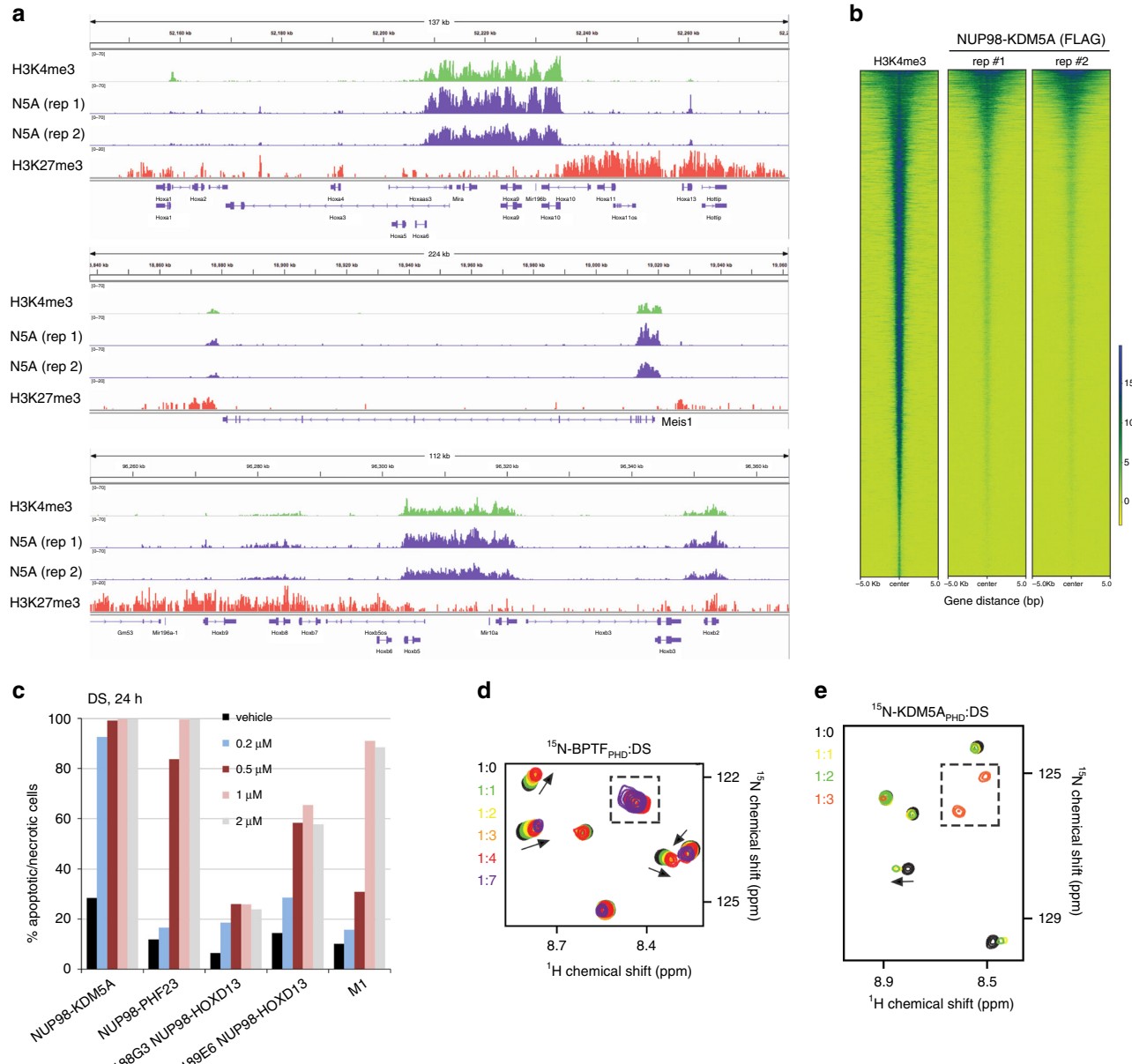

**Fig. 6 NUP98–KDM5A is enriched at H3K4me3 regions and is sensitive to DS. a** ChIP-seq profiles of H3K4me3, NUP98–KDM5A (N5A, two replicates), and H3K27me3 at the Hoxa, Meis1, and Hoxb loci in murine leukemia cells transformed by NUP98–KDM5A. **b** Heatmap of H3K4me3 and two replicas of NUP98–KDM5A ChIP-seq signals centered on H3K4me3-binding sites in a ±5.0-kb window in murine leukemia cells expressing FLAG-tagged NUP98–KDM5A. ChIP-seq signals were normalized to input. **c** DS induces cell death of NUP98–KDM5A expressing and NUP98–PHF23 expressing AML cell lines at earlier time points and lower DS concentrations than the NUP98–HOXD13 expressing cell line, confirming our previous results[9]. $n = 1$, source data are provided in the Source Data file. Superimposed $^{1}$H,$^{15}$N HSQC spectra of BPTF$_{PHD}$ **d** and KDM5A$_{PHD}$ **e** collected upon titration with DS. Spectra are color coded according to the protein:compound molar ratio. Dashed box indicates newly appeared cross-peaks.

**Concluding remarks and future directions.** Aberrations in histone methylation-dependent cellular processes have emerged as a common oncogenic pathway in human AML. Particularly, chromosomal translocations NUP98–PHF23, NUP98–KDM5A, and NUP98–BPTF define a subset of AMLs that are characterized by abnormal expression of Hoxa/b genes and Meis1. AMLs with such stemness-related gene signatures are highly lethal, underscoring an urgent need for the development of effective and specific anti-AML therapies. Our finding that NUP98–KDM5A directly binds to H3K4me3-marked "stemness" genes is consistent with previously observed occupancy of NUP98–PHF23[9] and suggests a common mechanism of action for NUP98 chimeras fused with the PHD finger-containing regions of KDM5A,

PHF23, and BPTF (Fig. 1b). Because disruption of the H3K4me3-binding sites of the PHD fingers in these NUP98 fusions suppresses oncogenic transformation, the strategy to develop inhibitors of the PHD finger has a strong rationale. Importantly, targeting this H3K4me3-specific reader could also be beneficial for coinciding blocking of all three oncogenic proteins. Histone readers are increasingly considered to be druggable, and a number of potent inhibitors of acetyllysine recognizing bromo-domain and YEATS domain have been developed in the past 10 years[24–27]. The pioneering studies[8,9,22] uncovering the pivotal role of the histone binding by the PHD finger of KDM5A, PHF23, and BPTF in NUP98 fusions in leukemic transformation and the feasibility of targeting these fusions by small molecules together

with our mechanistic findings further suggest that impairing the PHD finger function and/or structure may provide an effective way to treat AMLs associated with these NUP98 fusions.

## Methods

**Protein expression and purification**. The human PHF23 PHD domain (aa 338–393) was cloned into a pGEX 6p-1 vector. Proteins were expressed in BL21 (DE3) RIL in LB or minimal media supplemented with $^{15}NH_4Cl$, $^{13}C$-glucose and 0.05 mM $ZnCl_2$. Protein expression was induced with 0.2 mM IPTG for 16 h at 16 °C. The GST-tagged proteins were purified on glutathione Sepharose 4B beads (GE Healthcare) in 20 mM Tris-HCl (pH 7.0) buffer, supplemented with 150 mM NaCl, and 5 mM DTT. The GST tag was cleaved overnight at 4 °C with PreScission or Thrombin protease. Proteins were further purified by size exclusion chromatography and concentrated in Millipore concentrators. All mutants were generated by site-directed mutagenesis using the Stratagene QuikChange mutagenesis protocol, grown and purified as wild-type proteins. Primers are listed in Supplementary Table 2.

**NMR experiments**. NMR experiments were carried out at 298 K on Varian INOVA 500 and Bruker 600 MHz spectrometers. NMR samples contained 0.1 mM uniformly $^{15}N$-labeled WT or mutated PHD in PBS buffer (pH 6.5), 1 mM TCEP, and 8% $D_2O$. Binding was characterized by monitoring chemical shift changes in the proteins induced by histone H3 peptides (synthesized by SynPeptide). The $K_d$ values were determined by a nonlinear least-squares analysis in Kaleidagraph using the following equation:

$$\Delta\delta = \Delta\delta_{max}\frac{\left(([L]+[P]+K_d) - \sqrt{([L]+[P]+K_d)^2 - 4([P][L])}\right)}{2[P]}, \quad (1)$$

where $[L]$ is concentration of the peptide, $[P]$ is the concentration of the protein, $\Delta\delta$ is the observed chemical shift change, and $\Delta\delta_{max}$ is the normalized chemical shift change at saturation. Normalized chemical shift changes were calculated using the equation as follows:

$$\Delta\delta = \sqrt{(\Delta\delta H)^2 + (\Delta\delta N/5)^2}, \quad (2)$$

where $\Delta\delta$ is the change in chemical shift in parts per million.

The chemical shift assignments were obtained by a set of triple resonance experiments with nonlinear sampling using a 1.7 mM $^{13}C/^{15}N$-labeled PHF23$_{PHD}$ sample in 20 mM Tris (pH 7.0) buffer supplemented with 150 mM NaCl, 5 mM DTT, and 8% $D_2O$ as described[28]. Spectra were processed and analyzed with NMRPipe and CcpNmr Suite. Sequence-specific assignments were obtained using CcpNMR Analysis v2.1 and validated in the I-PINE web server.

For NMR titration experiments with inhibitors, 0.1 mM uniformly $^{15}N$-labeled PHF23$_{PHD}$, BPTF$_{PHD}$, or KDM5A$_{PHD}$ was used. BPTF$_{PHD}$ (aa 225–230) and KDM5A$_{PHD}$ (aa 1609–1659) were expressed and purified similar to PHF23$_{PHD}$. All experiments with AD were carried out in 20 mM Tris-HCl (pH 7.0) buffer, supplemented with 150 mM NaCl, 5 mM DTT, and 8% D2O. A concentrated stock of AD (100 mM) was made in methanol and pH was adjusted to 7.0. For experiments with DS, DTT was removed from protein samples by buffer exchange. A concentrated stock of DS (25 mM) was made in DMSO. All histograms were generated by calculating the normalized chemical shift change per residue between apo and ligand-bound protein states.

**X-ray crystallography**. Purified PHF23$_{PHD}$ was concentrated to 6–8 mg/ml in a buffer containing 20 mM Tris-HCl (pH 7.4) and 1 mM TCEP. Initial crystallization trials were set using the sitting-drop vapor-diffusion method. Good quality diffracting crystals were obtained at 18 °C by hanging drop vapor diffusion in 0.1 M Tris pH 8.65 and 25% tert-butanol. The data set was collected to 2.9 Å using the University of Colorado Denver Rigaku X-ray homesource. The phases were determined by molecular replacement in the CCP4 suite Phaser program using the PDB model 3O70. Model building was performed using Coot v0.8.2[29], and the structure was refined using Phenix Refine v1.6[30]. The final structure was verified by MOLProbity[31]. The X-ray diffraction and structure refinement statistics are summarized in Supplementary Table S1.

**Peptide pulldown assay**. For the peptide pulldown assays, 1 μg of biotinylated histone peptides with different modifications was incubated with 1.5 μg of GST-fused proteins in binding buffer (50 mM Tris-HCl 7.5, 250 mM NaCl, 0.1% NP-40) overnight. Streptavidin beads (Amersham) were added to the mixture, and the mixture was incubated for 1 h with rotation. The beads were then washed three times and analyzed using SDS-PAGE and western blotting using anti-GST (sc-459) at 1/2000 dilution.

**Fluorescence spectroscopy**. Spectra were recorded at 25 °C on a Fluoromax-3 spectrofluorometer (HORIBA). The samples containing 2.0 μM PHF23$_{PHD}$ fragment and progressively increasing concentrations of the peptide were excited at 295 nm. Experiments were performed in PBS buffer (pH 6.5) supplemented with

1 mM TCEP. Emission spectra were recorded over a range of wavelengths between 320 and 380 nm with a 1 nm step size and a 1 s integration time and averaged over three scans. The $K_d$ values were determined using a nonlinear least-squares analysis and the equation as described[32]:

$$\Delta I = \Delta I_{max}\frac{\left(([L]+[P]+K_d) - \sqrt{([L]+[P]+K_d)^2 - 4([P][L])}\right)}{2[P]}, \quad (3)$$

where $[L]$ is the concentration of the peptide, $[P]$ is the protein concentration, $\Delta I$ is the observed change of signal intensity, and $\Delta I_{max}$ is the difference in signal intensity of the free and bound states of the PHD domain. The $K_d$ value was averaged over three separate experiments, with error calculated as the standard deviation between the runs.

**SDS-PAGE electrophoresis**. Samples containing 0.1 mM PHD finger proteins were incubated with DS for 10 min, 30 min, 1 h, 3 h, and overnight. The reactions were quenched by flash-freezing at indicated time points, and the samples were stored at −20 °C. After collecting all samples, frozen proteins were thawed and analyzed immediately by SDS-PAGE under nonreducing and reducing (10 mM β-ME) conditions. 10 μl of loading buffer (containing SDS, EDTA, glycerol, TRIS-HCl, and bromophenol blue) was added to 10 μl of protein samples. The samples were then loaded onto a 15% polyacrylamide gel and resolved. The gels were stained with Coomassie blue.

**Cells and cell lines**. Immortal, cytokine independent leukemic cell lines that express a NUP98–PHF23 fusion (748T and 961C) were established from NUP98–PHF23 transgenic mice[9]. 748T is a precursor T-cell line, and 961C is a myeloid cell line. 188G3 and 189E6 are IL-3-dependent cell lines established from embryonic stem cells that express a NUP98–HOXD13 fusion from the endogenous NUP98 locus[33]. 7298/2 is a precursor T-cell line established from SCL/LMO1 double transgenic mice[34]. 32D is an IL-3 dependent, spontaneously immortalized, murine myeloid cell line[35]. Murine AML lines transformed by NUP98–KDM5A were generated and maintained as previously described[8]. All cell lines were maintained in Iscove's modified Dulbecco's medium supplemented with 15–20% fetal bovine serum (FBS), 100 mM L-glutamine and 100 μg/mL penicillin/streptomycin (Invitrogen). 188G3, 189E6, and 32D cells were also supplemented with IL-3 (10 ng/mL) (Peprotech).

**Apoptosis assay**. Apoptosis assays were conducted with an AnnexinV-FITC Apoptosis Detection Kit I (BD Pharmingen) using the manufacturer's recommended protocol and a FACSan flow cytometer (Cytek Biosciences). Experiments were carried out as previously described[9].

**Primary murine hematopoietic stem/progenitor cells (HSPCs)**. Primary bone marrow cells are harvested from femur and tibia of wild-type balb/C mice and then subject to a lineage-negative (Lin⁻) enrichment protocol to remove differentiated cell populations as described[8]. Such Lin⁻ enriched HSPCs were first stimulated in the base medium (OptiMEM, Invitrogen, cat#31985), 15% of FBS (Invitrogen, cat#16000-044), 1% of antibiotics, and 50 μM of β-mercaptoethanol complemented with a cytokine cocktail that contains 10 ng/mL each of murine SCF (Peprotech), Flt3 ligand (Flt3L; Sigma), IL-3 (Peprotech), and IL6 (Peprotech) for 4 days. MSCV-based retrovirus encoding the NUP98-PHD finger fusion oncogene was produced in 293 cells. Two days postinfection with retrovirus, murine HSPCs were subject to drug selection and then plated for assaying proliferation and differentiation in the same liquid base medium with SCF alone as described[8]. These in vitro cultured HSPC cells were routinely monitored under microscopy and cellular morphology examined by Wright–Giemsa staining.

**Chromatin immunoprecipitation followed by sequencing**. ChIP-seq was carried out and ChIP-seq data alignment, filtration, peak calling and assignment, and cross-sample comparison were performed as described[36]. Mouse anti-Flag tag M2 (Sigma F1804, 5 μg), anti-H3K3me3 (ab9050, 5 μg), and anti-H3K27me3 (Millipore 07-449, 5 μg) were used.

**Reporting summary**. Further information on research design is available in the Nature Research Reporting Summary linked to this article.

## Data availability

Coordinates and structure factors for PHF23$_{PHD}$ have been deposited in the Protein Data Bank with accession code PDB 6WXK (https://doi.org/10.2210/pdb6WXK/pdb). The ChIP-seq data are submitted to GEO under the accession number GSE146693. All other relevant data supporting the key findings of this study are available within the article and its Supplementary Information files or from the corresponding authors upon reasonable request. The source data underlying Figs. 1d, f, 2b, d, 3d–f, 4b, c, 5f, and 6c are provided in the Source Data file.

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

## Acknowledgements

This work was supported by the grants from NIH GM135671 and HL151334 to T.G.K., CA215284 to G.G.W., CA204020 to X.S. and by the Intramural Research Program of the National Cancer Institute, NIH ZIA SC0130378 and SC010379 to P.D.A. G.G.W. is an American Cancer Society Research Scholar and a Leukemia and Lymphoma Society Scholar. X.S. is a Leukemia & Lymphoma Society Career Development Program Scholar. Y.Z. is supported by NIH K99CA241301.

## Author contributions

Y.Z., Y.G., S.M.G., J.Z., K.R.V., K.L. and L.C. performed experiments and together with X.S., P.D.A., G.G.W. and T.G.K. analyzed the data. Y.Z., P.D.A., G.G.W. and T.G.K. wrote the paper with input from all authors.

## Competing interests

The authors declare no competing interests.
