## [Peer Review File · Nature Communications]

Reviewer #1 (Remarks to the Author):

Zhang et al report in vitro and cellular analysis of oncogenic NUP98 fusions with PHD fingers from PHF23, KDM5A and BPTF. They claim that inhibition of the PHD finger H3K4me3 interaction by FDA-approved drugs disulfiram and amiodarone is the mechanism of observed cell death in cancer cell lines.

Below I will specify my thoughts on the data, broken down by figure. Given the global crisis I think it would be unfair to ask for more experiments so I have taken that into consideration, and simply given my assessment of the merits of the data as they stand now:

Fig1: The NMR analysis of H3 peptide binding is clear and well done. It's not surprising that the PHD finger selects for H3K4me3 but the data are convincing. As acknowledged by the authors, Wang et al. Nature (2009) reaches the same conclusion with pulldown assays. The NMR data corroborate those findings. Fig 1f is a similar pulldown to the 2009 paper and it's reassuring (!) that the authors can reproduce the 2009 data.

Fig2: The authors provide an interesting structure of the PHF23 PHD where they see that - despite H3K4me3 peptide being present in the crystallization drops - the PHD finger forms crystallographic dimers with the N-terminal part of the construct mimicking the H3 tail. They therefore turn to NMR to determine the H3K4me3 binding site. The binding site appears to be another example of a well-characterized hydrophobic cage binding mode, as expected based on numerous other structures of PHD fingers in complex with H3K4me3 (around 30 structures based on a quick PDB search). Again, the NMR is done well and the crystal structure appears of reasonable quality given the relatively low resolution (2.9Å). I wonder if the crystallographic dimer is physiologically relevant.

Fig3: The authors build on previous data by trying to get at the mechanism of cancer cell growth inhibition by amiodarone (AM) and disulfiram (DS). As far as I can tell, panel d-f recapitulates the data shown in a previous paper from the same authors (Gough et al 2014). They hypothesised previously that these compounds specifically inhibit the PHD-H3K4me3 interaction. Now, in panels b and c the authors use NMR chemical shift perturbations (CSPs) to show interactions of these small molecules with the PHD finger. These interactions are rather unconvincing (very minor CSPs, not beyond the linewidth of the affected peaks). This type of minor CSP is often caused by small pH changes or solvent/buffer mismatch effects - so controls should be shown to rule this out (if they have them), or caveats clearly stated in the text. No K_d is determined, presumably because at high concentrations the compounds unfold or crosslink the protein (panel 3i), making the NMR difficult. Nonetheless I find it hard to assess the specificity of the interactions and inhibition, and you could argue the paper rather hinges on this. It's a bit peculiar that the molecules have completely different structures but target the same site (panels g, h) - this could point again to solvent effects, or non-specific hydrophobic interactions with the hydrophobic H3K4me3 binding pocket. It seems unlikely (to me at least) that the crosslinking (DS) or unfolding (AM) behaviour at ~300µM and 2mM respectively could really be so specific for these proteins. There are many other cellular proteins with labile Zn and/or reactive Cys and/or hydrophobic surfaces. The molecules are FDA approved (which is good) - but a quick search suggests that they have several other putative cellular targets. There may be a correlation between crosslinking of the PHD finger by DS in vitro and cancer cell death, but I don't think this proves causation...

Fig4: The downregulation of PHF23 targets in the presence of DS (panel b) is interesting and indeed striking, but as far as I can tell this result was already shown by the authors before (Gough et al 2014)? Although all the shown mRNA levels are downregulated (normalised to 18S rRNA), no controls (e.g. a housekeeper gene) are shown so it's hard again to assess specificity.

Fig5: Here the authors show that mutation of NUP98-BPTF in the H3K4me3 binding residues abrogates oncogenic transformation, proving that H3K4me3 binding is necessary. This is nice and quite convincing. As acknowledged by the authors, this was done already for other fusions in the

2009 paper, so the data aren't hugely surprising. Nonetheless it's good to corroborate and strengthen previous findings.

Fig6: ChIP-seq analysis shows very nicely that the NUP98-KDM5a fusion colocalizes with H3K4me3 as expected. The data seem of high quality. This mirrors the same experiments done by these authors (the 2014 paper) with the NUP98-PHF23 fusion. I think panel c shows that DS does still kill NUP98-HOXD13 expressing AML cell lines, just not as potently as the other lines - in contrast to what is stated in the figure legend?

Comments on stats/errors:

- Mostly OK, but 3d-f, 4c and 6c are lacking replicates/error bars/confidence intervals.
- It's great that the ChIP-seq replicates are shown in Fig 6.

Overall, most of the experiments (NMR, crystallography, ChIP-seq) seem well done. I'm not an expert in this specific cancer pathway or the cell death assays used.

Unfortunately, while (mostly) sound from a technical point of view, I can't quite see that the paper will substantially influence thinking in the field, simply because most of the biological results/pathways/conclusions appear to have been shown already (in another form, or closely related form) in previous studies - most notably, Wang et al Nature 2009; Gough et al Cancer Discovery 2014. Specific binding modes of the FDA-approved small molecules (that therefore could be repurposed as leukemia drugs) would be exciting. Although a valiant effort has been made to show binding with NMR titrations, those in vitro results are probably the least convincing in the paper. Lots more experiments would be needed to firm up the mechanism and a targeted mode of action, but I think it's unreasonable to ask for these at a time when people should be at home and not in the lab. I think therefore that the conclusions regarding DS and AM modes of action need to be toned down substantially - but in doing so, I guess that might limit the wider interest somewhat?

--

Christopher Douse, March 2020

Reviewer #2 (Remarks to the Author):

The manuscript by Kutateladze and colleagues describes the molecular basis by which the NUP98-PHF23 PHD domain recognizes trimethyl-Lys4 in histone H3 (H3K4me3) and how small molecule compounds disrupt this interaction in cancer cell-based assays. A subset of acute myeloid leukemia (AML) cases involves gene translocations of nucleoporin 98 (NUP98) fused to epigenetic factors that harbor H3K4me3-binding PHD domains, including PHF23, BPTF, and KDM5A. The PHD domains of these NUP98 fusions recognize H3K4me3 in chromatin, stimulating transcription. To gain insight into this recognition, the authors used structural and biochemical approaches to characterize the interactions between the PHF23 PHD domain and H3K4me3. The structure of the PHF23 PHD domain reveals an aromatic cage similar to cages identified in BPTF and KDM5A that bind K4me3 in histone H3. To further probe this interaction, they examined whether the small molecule inhibitors amidarone (AD) and disulfiram (DS) associate with the PHF23 PHD domain and found that they bind in the vicinity of the H3K4me3 binding cleft. Furthermore, DS treatment resulted in disulfide bonded oligomerization of the PHF23 PHD domain. Similar results were obtained with DS and the PHD domains of BPTF and KDM5A. Finally, the authors showed that the NUP98 fusions with the PHD domain-containing proteins led to leukemic transformation and that DS treatment induced cell death in AML cell lines harboring the NUP98 fusions. Together, this work furnishes important new insight into the role of NUP98-PHF23 and other PHD domain protein fusions in leukemogenesis and is recommended for publication. The following points should be addressed prior to publication.

- 1) In the SDS-PAGE analysis of DS and the PHF23, BPTF, and KDM5A PHD domains (Figure 3i and Supp. Figure 8), did the SDS sample loading buffer contain EDTA? If so, the EDTA would chelate the Zn(II) ions in the PHD domains, exposing the Zn-coordinating cysteines and potentially enhancing disulfide bond formation. This point should be addressed in the revised manuscript.
- 2) Related to #1, does reaction of DS with a Zn-coordinating cysteine result in loss of the Zn(II) from the PHD domains of PHF23, BPTF, and KDM5A?
- 3) Is it known whether DS causes disulfide bonded oligomerization of other PHD domains, such as those that recognize unmodified H3K4? How specific is DS for H3K4me-recognizing PHD domains?
- 4) Please provide a legend in Figure 1d describing the black, beige and purple data points and fitted curves for the fluorescence binding data.
- 5) The manuscript describes the crystal structure of the PHF23 PHD domain in which the N-terminal residues DLIT (residues 338-341) from one PHF23 PHD molecule bind in the H3K4me binding cleft of a neighboring PHF23 PHD molecule. Figure 2b illustrates the binding of Ile340 in the aromatic cage, but the interactions with the other residues in the DLIT sequence are not depicted. An additional figure illustrating the interactions between the DLIT residues and the PHF23 PHD should be included in the manuscript or supplementary information.
- 6) The RMSD values for the alignment of the structures of the PHF23, BPTF, and KDM5A PHD domains in Figure 5c should be reported.

Reviewer #3 (Remarks to the Author):

In this manuscript, the authors reported new mechanistic insights into the molecular functions of the Phd fingers of oncofusion proteins NUP98-PHF23, NUP98-KDM5A and NUP98-BPTF in acute myeloid leukemia (AML). Specifically, the authors demonstrated that binding of by these Phd fingers to histone H3 trimethylated lysine 4 (H3K4me3), a histone mark for gene transcriptional activation, is a common mechanism by which these oncofusion proteins function to promote leukemogenesis. Small molecule compounds that target the Phd finger binding to H3K4me3 can block the oncofusion's chromatin engagement and its contributions to transcriptional activation of oncogenes required for leukemogenesis. Thus, this study established a direct role of NUP98-associated oncofusion proteins in leukemic transformation through their Phd finger's recognition of H3K4me3 in chromatin. Overall, this is an elegant study with important findings, executed with combined use of structural biology and chromatin genomics techniques, which together enabled the authors to demonstrate that the Phd fingers, a major class of chromatin readers, likely represent a new class of anti-cancer drug targets, particularly for acute myeloid leukemia. Therefore, with addressing several comments listed below, this work is suitable for publication in Nature Comm.

Specific comments:

1. Figure 3, the molar concentrations of the proteins as well as small molecule compounds used in the NMR titration should be provided.
2. Figure 3d, the chemical inhibitor AD (amidarone) exerted a rather sharp transition of affecting cell viability from 72 to 96 hours. An explanation for this observation will be needed.
3. Molecular weight markers are missing for some western blots such as Figure 3i, Figure 5e, s-Figure 4, s-Figure 8.

We thank the Reviewers for the insightful and very constructive comments, which were helpful in revising and strengthening this manuscript.

Reviewer 1, Comment 1: *Below I will specify my thoughts on the data, broken down by figure. Given the global crisis I think it would be unfair to ask for more experiments so I have taken that into consideration, and simply given my assessment... - we thank this reviewer and other reviewers for support and understanding.*

... Fig2: The authors provide an interesting structure of the PHF23 PHD ... the NMR is done well and the crystal structure appears of reasonable quality... I wonder if the crystallographic dimer is physiologically relevant.

Author's response: we believe it is unlikely that the dimer is physiologically relevant - the serine residue derived from the vector is involved in contact with the PHD finger.

Reviewer 1, Comment 2: *Fig3: in panels b and c the authors use NMR chemical shift perturbations (CSPs) to show interactions of these small molecules with the PHD finger. These interactions are rather unconvincing (very minor CSPs, not beyond the linewidth of the affected peaks). This type of minor CSP is often caused by small pH changes or solvent/buffer mismatch effects - so controls should be shown to rule this out (if they have them), or caveats clearly stated in the text.*

Author's response: although the CSPs are small (more pronounced for the PHD fingers of BPTF and KDM5A in Fig. 6d, e), they are significant and not uniform, indicating that resonances exhibit unique changes in local chemical environments.

Furthermore, Wagner et al. (ref. 22) have shown that AD inhibits binding of H3K4me3-specific domains, including PHDs of KDM5A and RAG2. DS acts on PHD fingers of AIRE and BHC80 but not of RAG2.

The following sentence has been added on page 7: These results were in agreement with the reports demonstrating that AD and its analogues display inhibitory activity toward H3K4me3-binding domains, whereas some PHD fingers are sensitive to DS²².

It's a bit peculiar that the molecules have completely different structures but target the same site (panels g, h) - this could point again to solvent effects, or non-specific hydrophobic interactions with the hydrophobic H3K4me3 binding pocket. It seems unlikely (to me at least) that the crosslinking (DS) or unfolding (AM) behaviour at ~300uM and 2mM respectively could really be so specific for these proteins. There are many other cellular proteins with labile Zn and/or reactive Cys and/or hydrophobic surfaces. The molecules are FDA approved (which is good) - but a quick search suggests that they have several other putative cellular targets....

Author's response: these compounds were chosen specifically because they mimic methyllysine, and as we predicted, some residues in the methyllysine-binding site were perturbed by either AD or DS. However, the patterns of CSPs induced by these compounds, as expected, were distinctly different.

There are other targets of AD and DS, including those for which these drugs were originally developed – mitochondrial aldehyde dehydrogenase ALDH2 in 1982 and HERG human cardiac K⁺ channel in 1999.

Please also see our response to Comment 3 of Reviewer 2.

Reviewer 1, Comment 3: Fig4: The downregulation of PHF23 targets in the presence of DS (panel b) is interesting and indeed striking, ... Although all the shown mRNA levels are downregulated (normalised to 18S rRNA), no controls (e.g. a housekeeper gene) are shown so it's hard again to assess specificity.

Author's response: we examined the genes that have previously been identified as targets for NUP98 fusions. The following sentence has been added in Fig. 4c legend: "qRT-PCR quantification of previously identified⁹ target genes normalized to 18S RNA as internal control confirms downregulation of Hoxa5, Hoxa7, Hoxa9, Hoxa10, Hoxb5 and Meis1 in the presence of DS."

Reviewer 1, Comment 4: Fig6: ... I think panel c shows that DS does still kill NUP98-HOXD13 expressing AML cell lines, just not as potently as the other lines - in contrast to what is stated in the figure legend?

Author's response: the Fig. 6c legend has been revised to: "DS induces cell death of NUP98-KDM5A expressing and NUP98-PHF23 expressing AML cell lines at earlier time points and lower DS concentrations than the NUP98-HOXD13 expressing cell line..."

Reviewer 1, Comment 5: Comments on stats/errors:

-Mostly OK, but 3d-f, 4c and 6c are lacking replicates/error bars/confidence intervals.

Author's response: "For additional trials see^{9m}" and "...confirming our previous results^{9m}" have been added in Figs. 3d-e, 3f and 6c legends.

Reviewer 2, Comment 1: ... In the SDS-PAGE analysis of DS and the PHF23, BPTF, and KDM5A PHD domains (Figure 3i and Supp. Figure 8), did the SDS sample loading buffer contain EDTA? If so, the EDTA would chelate the Zn(II) ions in the PHD domains, exposing the Zn-coordinating cysteines and potentially enhancing disulfide bond formation. This point should be addressed in the revised manuscript...

Author's response: we have performed experiments using SDS loading buffer without EDTA (figure on the left) and with EDTA (in the manuscript) and obtained similar results (indicating that such a small amount of EDTA present during a short period of time does not produce a detectable effect). In both cases, EDTA was absent in running buffer.

Reviewer 2, Comment 2: Related to #1, does reaction of DS with a Zn-coordinating cysteine result in loss of the Zn(II) from the PHD domains of PHF23, BPTF, and KDM5A?

Author's response: although we did not measure the release of zinc, previous studies have shown the release of zinc upon treatment of the PHD finger of KDM5A with DS (ref. 22).

Reviewer 2, Comment 3: Is it known whether DS causes disulfide bonded oligomerization of other PHD domains, such as those that recognize unmodified H3K4? How specific is DS for H3K4me-recognizing PHD domains?

Author's response: Wagner et al. (ref. 22) have reported that upon addition of DS, zinc is released from the PHD fingers of BHC80 and AIRE (which recognize unmodified H3K4), but BHC80 and AIRE are not translocation partners of NUP98. It is unlikely that binding of DS is specific, though DS does not cause the release of zinc from the RAG2 PHD finger (ref. 22).

Reviewer 2, Comment 4: *Please provide a legend in Figure 1d describing the black, beige and purple data points and fitted curves for the fluorescence binding data.* – we have added this description.

Reviewer 2, Comment 5: *...Figure 2b illustrates the binding of Ile340 in the aromatic cage, but the interactions with the other residues in the DLIT sequence are not depicted. An additional figure illustrating the interactions between the DLIT residues and the PHF23 PHD should be included in the manuscript or supplementary information.* – as suggested, we have added new Suppl. Fig. 1.

Reviewer 2, Comment 6: *The RMSD values for the alignment of the structures of the PHF23, BPTF, and KDM5A PHD domains in Figure 5c should be reported.* – we have included RMSDs in Fig. 5c legend.

Reviewer 3, Comment 1: *Figure 3, the molar concentrations of the proteins as well as small molecule compounds used in the NMR titration should be provided.* – this info has been added in Fig. 3b, c legend.

Reviewer 3, Comment 2: *Figure 3d, the chemical inhibitor AD (amiodarone) exerted a rather sharp transition of affecting cell viability from 72 to 96 hours. An explanation for this observation will be needed.* – a possible explanation could be that AD affects an apoptosis set point (961C cell undergo spontaneous apoptotic cell death at a certain cell concentration), although this idea needs to be at least initially tested.

Reviewer 3, Comment 3: *Molecular weight markers are missing for some western blots such as Figure 3i, Figure 5e, s-Figure 4, s-Figure 8.* – we have added MW markers in these figures.